# Evaluation of Antimicrobial Activity by Marine *Nocardiopsis dassonvillei* against Foodborne *Listeria monocytogenes* and Shiga Toxin-Producing *Escherichia coli*

**DOI:** 10.3390/microorganisms11102539

**Published:** 2023-10-12

**Authors:** Siyanda S. Ngema, Solomuzi H. Khumalo, Michael C. Ojo, Ofentse J. Pooe, Tsolanku S. Malilehe, Albertus K. Basson, Evelyn Madoroba

**Affiliations:** 1Department of Biochemistry and Microbiology, University of Zululand, Private Bag X1001, KwaDlangezwa 3886, South Africa; ngemasiyanda05@gmail.com (S.S.N.); elsherawy7@gmail.com (S.H.K.); mikekonyegwachie2015@gmail.com (M.C.O.); bassona@unizulu.ac.za (A.K.B.); 2Discipline of Biochemistry, School of Life Sciences, University of KwaZulu-Natal, Private Bag X54001, Durban 4000, South Africa; pooeo@ukzn.ac.za; 3Department of Water and Sanitation, University of Limpopo, Private Bag X1106, Polokwane 0727, South Africa; tsolanku.maliehe@ul.ac.za

**Keywords:** marine actinomycetes, *Nocardiopsis* sp., antibacterial potential, molecular docking, *Listeria* sp.

## Abstract

The emergence of multidrug-resistant pathogens creates public health challenges, prompting a continuous search for effective novel antimicrobials. This study aimed to isolate marine actinomycetes from South Africa, evaluate their in vitro antimicrobial activity against *Listeria monocytogenes* and Shiga toxin-producing *Escherichia coli*, and characterize their mechanisms of action. Marine actinomycetes were isolated and identified by 16S rRNA sequencing. Gas chromatography–mass spectrometry (GC–MS) was used to identify the chemical constituents of bioactive actinomycetes’ secondary metabolites. Antibacterial activity of the secondary metabolites was assessed by the broth microdilution method, and their mode of actions were predicted using computational docking. While five strains showed antibacterial activity during primary screening, only *Nocardiopsis dassonvillei* strain SOD(B)ST2SA2 exhibited activity during secondary screening for antibacterial activity. GC–MS identified five major bioactive compounds: 1-octadecene, diethyl phthalate, pentadecanoic acid, 6-octadecenoic acid, and trifluoroacetoxy hexadecane. SOD(B)ST2SA2′s extract demonstrated minimum inhibitory concentration and minimum bactericidal concentration, ranging from 0.78–25 mg/mL and 3.13 to > 25 mg/mL, respectively. Diethyl phthalate displayed the lowest bacterial protein-binding energies (kcal/mol): −7.2, dihydrofolate reductase; −6.0, DNA gyrase B; and −5.8, D-alanine:D-alanine ligase. Thus, marine *N. dassonvillei* SOD(B)ST2SA2 is a potentially good source of antibacterial compounds that can be used to control STEC and *Listeria monocytogenes*.

## 1. Introduction

The 21st century is plagued by the emergence of multidrug-resistant bacteria such as foodborne pathogenic strains of *Listeria* and Shiga toxin-producing *Escherichia coli* (STEC) [1,2]. The major contributing factors to their antimicrobial resistance include the routine application of antimicrobials in domestic livestock for growth promotion, disease treatment, and prophylaxis [3]. These pathogens acquire resistance towards antibiotic effects mainly through gene mutation, acquisition of resistant genes through horizontal gene transfer, and biofilm formation [4]. Antimicrobial resistance (AMR) is anticipated to lead to approximately 10 million deaths annually by 2050, and the World Health Organization report estimates that, 700,000 global fatalities could be linked to AMR each year [5]. Therefore, the increase and spread of antimicrobial resistance pose a danger to public health. AMR further threatens to push the health sector to a pre-antimicrobial era [6].

*Listeria* spp. and STEC both cause diarrhea in humans, which may progress to life-threatening conditions in the vulnerable groups [7]. Infection with *Listeria monocytogenes* could result in listeriosis, which may proceed to meningitis and encephalitis in immune-compromised individuals [8], and fetal malformations, stillbirths and spontaneous abortions in pregnant women [9]. Worldwide, it is estimated that, each year, there are 23,150 cases and 5463 deaths from listeriosis [10,11]. Moreover, listeriosis has the third-highest case mortality rate (up to 30%) among the foodborne diseases [12]. Currently, *Listeria* contains 27 species [13], but only four of them are commonly isolated from food, namely: *L. welshimeri*, *L. seeligeri*, *L. monocytogenes*, and *L. ivanovii* [14]. Among them, *L. monocytogenes* is the only species that is a major human pathogen of public health significance [15,16]. Rarely, *L. ivanovii*, *L. innocua*, and *L. seeligeri* cause human infections [17]. *L. ivanovii* principally causes the disease in ruminants [18].

Infection with STEC may deteriorate into hemolytic uremic syndrome (HUS) in 5 to 10% of the cases, especially children and the elderly [19]. The syndrome is characterized by hemolytic anemia, thrombocytopenia, and nephropathy [20]. Worldwide it is believed that approximately 2,801,000 cases can be attributed to STEC-induced acute illnesses, while 3890 cases are HUS, and 230 deaths annually [21,22]. Based on the flagella (H) and somatic (O) antigens, STEC is divided into over 600 serogroups, and all of them have the ability to produce Shiga toxins. However, due to their low infectious dose (10 to 100 cells), the following STEC serogroups are commonly linked to severe disease in humans: O26, O71, O103, O111, O121, O145, and O157 [1,23,24]. Among them, serogroup O157 is most frequently associated with serious food poisoning [23]. Recently, the incidence of serious disease due to non-O157 serogroups has been increasing [24]. The treatment of listeriosis is antimicrobial chemotherapy [17]. However, for STEC infections, the use of antibiotics is a debatable matter, and generally the use of antibiotics is not recommended. Therefore, the disease is usually managed by symptomatic treatment. Nonetheless, antibiotic treatment with inhibitors of protein and cell wall synthesis can be an option when specific criteria regarding duration of disease, serotype, virulence profiles, and patient group are satisfied [25].

As multidrug resistance of the implicated bacteria elevates and worsens morbidity and mortality rates, novel antimicrobials are needed to combat this crisis [26]. The actinomycetes are Gram-positive bacteria and produce antimicrobials and other bioactive compounds [27]. Over 45% of the total discovered biologically active natural metabolites are from actinomycetes [28]. Furthermore, over 80% of the therapeutically useful antibiotics are from this bacterial group, with 50% of the antibiotics from the genus *Streptomyces* [29,30]. However, the possibility of finding novel metabolites with unique chemical structure from *Streptomyces* has significantly decreased, predominantly because of genetic exchange among species during evolution [31]. Therefore, bioprospecting of promising rare actinomycetes from less-explored environments has been given primary attention recently. These actinomycetes have been described as “strains” other than commonly known *Streptomyces* or strains with less frequency of isolation under conventional cultivation techniques [31]. The rare actinomycetes genera include, among others, *Frankia*, *Micromonospora*, *Micrococcus*, *Nocardia*, *Arthrobacter*, and *Nocardiopsis* [32].

*Nocardiopsis* species are biotechnologically important producers of different bioactive compounds (antimicrobials, anticancer agents, tumor inducers, toxins, immunomodulators, and others) and novel extracellular enzymes [33]. Ecologically, these metabolites aid the species to survive and thrive even in extreme habitats including marine, hypersaline habitats, desert regions, and salterns [34]. Members of this genus are Gram-positive aerobes, catalase-positive and non-acid-fast actinomycetes [35]. They possess nocardioform substrate mycelia, and their aerial mycelia contain long chains of spores, with high genomic guanine and cytosine content [36]. Variations in the properties of strains isolated in different environments and their bioactive compounds is undoubtably significant and cannot be ignored [34]. Furthermore, when compared to marine environments, terrestrial environments have been explored extensively [28]. Thus, the hugely underexploited marine environments may harbour new actinomycetes with unique bioactive secondary metabolites [37]. Mainly due to the harsh physicochemical conditions found in the marine environment (such as high salinity, high pressure and cold temperatures) tend to favour the microbial production of structurally and functionally unique molecules that may be of industrial and pharmacological importance [38]. To the best of our knowledge, there are no studies that have reported the antimicrobial potential of marine actinomycetes from Sodwana Beach, KwaZulu-Natal (KZN), South Africa, against *Listeria* species and STEC.

Therefore, the aim of this study was to investigate the antimicrobial production of actinomycetes from the selected beaches in KwaZulu-Natal Province, South Africa, for their antibacterial activity against *L. monocytogenes* and STEC isolates from beef. Moreover, the molecular interactions of the metabolites with the target receptor proteins were investigated to ascertain their mode of antibacterial activity.

## 2. Materials and Methods

### 2.1. Media Formulations

#### 2.1.1. Cross-Streak Agar

A medium comprising of yeast extract (3 g) (Biolab, Modderfontein, South Africa), peptone (3 g) (Biolab, Wadesville, South Africa), casein (3 g) (Sigma, Steinheim, North Rhine-Westphalia, Germany), starch (8 g) (Merck Pty Ltd., Modderfontein, South Africa), K_2_HPO_4_ (0.5 g) (Minema chemicals, Johannesburg, South Africa), MgSO_4_·7H_2_O (0.5 g) (Minema chemicals, Durban, South Africa), NaCl (2 g), agar (15 g) (Neogene, Heywood, UK), and distilled water (1000 mL) was used. The pH of the medium was adjusted to 7 using 0.1 M of HCl and 0.1 M of NaOH.

#### 2.1.2. Fermentation Broth

A broth medium containing yeast extract (3 g), peptone (3 g), casein (3 g), starch (8 g), K_2_HPO_4_ (0.5 g), MgSO_4_·7H_2_O (0.5 g), glycerol (3 g) (Merck, Pty, Gauteng, South Africa), CaCO_3_ (0.75 g) (Minema chemicals, Johannesburg, South Africa), and filtered marine water (1000 mL) was used. The pH of the medium was adjusted to 7 using 0.1 M of HCl and 0.1 M of NaOH.

### 2.2. Test Bacteria

The test bacteria used in this study were isolated and characterized as part of a project on “Antimicrobial Resistance Among Foodborne Pathogens” with reference number THRIP/22/30/11/2017 (Table 1). *L. monocytogenes* ATCC 15313 and *E. coli* O157:H7 ATCC 43888 were procured from the American Type Culture Collection (ATCC, Manassas, VA, USA) and used as controls.

### 2.3. Sample Collection

In March 2021, marine water was aseptically collected for analysis from six beaches of KwaZulu-Natal Province: Alkantstrand, Blythedale, Salt Rock, Sodwana, Mthunzini, and Tinley Manor (Figure 1) using a grab sampling method. In each beach, the water samples were collected every 100 m from 3 different consecutive sites. In every site, the samples were obtained in triplicate (2 L each) at a depth of 36 cm. The samples were then placed in cooler boxes and transported to the Microbiology laboratory of the University of Zululand.

### 2.4. Physicochemical Parameters of the Seawater

The following physicochemical parameters were measured at all beaches in triplicate in situ: dissolved oxygen (DO), pH, pressure, salinity, specific conductivity, temperature, and total dissolved solids (TDS) using a multiparameter water meter (Hanna HI 98194, Romania).

### 2.5. Sample Preparation

The water samples were immediately prepared and analyzed upon arrival. One half of the samples was treated by heating at the water bath at 50 °C for 1 h [39,40]. The other half was left untreated. Serial dilutions were then conducted up to 10^−1^. Subsequently, the water samples were concentrated through 0.45-micron nitrocellulose membrane filters using the EZ-stream^TM^ pump filtration system (Merck Millipore, Molsheim, France) in preparation for culture-based analysis.

### 2.6. Isolation of Actinomycetes

Following the filtration of the water samples, the filtration discs used were placed on three different culture media, namely, actinomycetes isolation agar (Condalob, Madrid, Spain), yeast mold agar (Condalob, Madrid, Spain), and marine agar (Mast Group, Merseyside, UK), which were then incubated for 31 days at different temperatures based on the physicochemical properties of water (Table 2). Thereafter, the presumed colonies of actinomycetes were selected and subcultured on actinomycetes isolation agar. They were then preserved in 30% glycerol at −80 °C for long-term storage.

### 2.7. The Screening for Antibacterial Compounds Production in Actinomycetes

The presumptive actinomycetes were screened for antibacterial activity against the test bacteria.

#### 2.7.1. Primary Screening of Antibacterial Activity

The isolates were screened for antibiotic production by the cross-streak method against the test bacteria. Briefly, the presumptive actinomycetes were streaked as a straight line in the middle of a cross-streak agar medium. The inoculated plates were subsequently incubated for 7 days at different temperatures depending on the physicochemical parameters shown in Table 2. Thereafter, the test bacterial strains at exponential phase were adjusted to 1 × 10^6^ colony-forming units per milliliter (CFU/mL) and streaked perpendicularly to the actinomycetes. The plates were then incubated at 37 °C for 24 h and observed for the formation of the inhibition zones [41,42].

#### 2.7.2. Secondary Screening for Antibacterial Activity of Actinomycete Isolates

The active actinomycetes, based on primary screening, were inoculated onto 100 mL of the fermentation broth separately. The inoculated broths were then incubated for 7 days at 160 rpm and at different temperatures based on the physicochemical parameters shown in Table 2. The culture broths were centrifuged for 30 min at 5000 rpm. Thereafter, antimicrobial activity was evaluated by the agar-well diffusion method. Briefly, the test bacterial inoculums, at the logarithm growth phase, were adjusted to 1 × 10^6^ CFU/mL. Subsequently, the test bacterial lawns were prepared on Mueller–Hinton agar plates, followed by the boring of the wells (6 mm diameter). The cell-free supernatant (100 µL) was pipetted into the wells and incubated at 37 °C for 24 h. The agar plates with uninoculated supernatant served as controls, and the zones of inhibition were recorded in milliliters [41,43].

### 2.8. Identification of Actinomycetes

The molecular identification of the presumptive isolates that exhibited antibacterial activity during the primary and secondary screenings of antibacterial activity was conducted by 16S rDNA gene sequencing. Briefly, genomic DNA was extracted from the cultures using the Quick-DNA^TM^ Fungal/Bacterial Miniprep Kit (Zymo Research, Catalogue No. D6005). The 16S target region was amplified using OneTaq^(R)^ Quick-Load^(R)^ 2X Master Mix (NEB, Catalogue No. M0486) with the primers 16S-27F (5′AGAGTTTGATCMTGGCTCAG’3) and 16S-1492R (5′CGGTTACCTTGTTACGACTT’3). The PCR cycling conditions used were: the initial denaturation step for 4 min at 94 °C, followed by denaturation (30 cycles at 94 °C for 30 s), annealing at 55 °C for 40 s and extension at 72 °C for 1 min, and a final extension for 7 min at 72 °C [44]. The PCR products were run on a gel and gel extracted with the Zymoclean^TM^ Gel DNA Recovery Kit (Zymo Research, Catalogue No. D4001). The extracted fragments were sequenced in the forward and reverse direction (Nimagen, BrilliantDye^TM^ Terminator Cycle Sequencing Kit V3.1, BRD3-100/1000) and purified (Zymo Research, ZR-96 DNA Sequencing Clean-up Kit^TM^, Catalogue No. D4050). The purified fragments were analyzed on the ABI 3500xl Genetic Analyzer (Applied Biosystems, ThermoFisher Scientific, Foster City, CA, USA) for each reaction for every sample. CLC Bio Main Workbench v7.6 was used to analyze the .ab1 files generated by the ABI 3500XL Genetic Analyzer, and the results were obtained by BLAST search (NCBI).

### 2.9. Phylogenetic Analysis

A phylogenetic tree of the strain showing antibacterial activity in both the primary and secondary screenings, *Nocardiopsis dassonvillei* strain SOD(B)ST2SA2, and its phylogenetic neighbours was constructed based on 16S rRNA sequences. The tree was constructed using MEGA 11 software, and the evolutionary history of SOD(B)ST2SA2 was inferred using the neighbor-joining (NJ) method. The NJ bootstrapping was performed with 1000 replicates.

### 2.10. Extraction of the Secondary Metabolites

*N. dassonvillei* SOD(B)ST2SA2 was inoculated into the fermentation medium and cultured for up to 7 days at 23 °C in a shaking incubator at 160 rpm. Following the incubation, the broth was centrifuged at 10,000 rpm for 15 min, and the supernatant was collected. To the collected supernatant, an equal volume of various solvents was added separately and kept in a rotary shaker (with a shaking speed of 160 rpm) at different times to extract the metabolites: 1 h for methanol, and 24 h for chloroform and ethyl acetate. After shaking, the mixtures were left to stand in separating funnels at room temperature for up to 24 h to separate the organic phase from the aqueous phase. Thereafter, the organic phase was concentrated by rotary vacuum evaporator [33,42,45]. The concentrated extracts were each weighed and dissolved in 10% Tween 80 and tested for their antibacterial activity against the field and ATCC strains of *L. monocytogenes* and STEC by agar-well diffusion method [46].

### 2.11. Chemical Composition Analysis of the N. dassonvillei SOD(B)ST2SA2′s Extract

Antibacterial compounds in the bioactive chloroform extract were identified by gas chromatography–mass spectroscopy (GC–MS). This was done by injecting 1 µL of sample into an RT x −5 column (30 × 0.32 nm) of the GC–MS model (Perkin Elmer, Clarus 500, Waltham, MA, USA). The carrier gas used was helium (3 mL/min). Identification of the chemical components of the extract was conducted using Perkin Elmer (Clarus 500, Waltham, MA, USA) gas chromatography coupled with (Clarus 500, Waltham, MA, USA) mass spectrometer (MS) [45].

### 2.12. Antibacterial Assay of the Chloroform Extract

#### 2.12.1. Minimum Inhibitory Concentration (MIC) of the Extract

To quantitatively estimate the minimum inhibitory concentration (MIC) of the extract, the 96-well microdilution method was used [47,48]. Briefly, Mueller–Hinton broth (50 μL) was pipetted into each well. Thereafter, 50 μL of the extract (100 mg/mL in 10% Tween 80) was poured into each well in the first row of the microplate and agitated thoroughly [49]. By successive dilutions, 50 μL of the extract mixture per well in row A was transferred into wells located down the column. To ensure that the volume in each well remains 50 μL, 50 μL of the mixture was emptied from each well of the last row of the microplate. Fifty microliters of the test bacteria, with a concentration diluted to 1 × 10^6^ CFU/mL, were pipetted into each well. Ciprofloxacin (2 × 10^−2^ mg/mL) and 10% Tween 80 were positive and negative controls, respectively. The microplate was covered and placed in an incubator (MODEL) overnight at 37 °C. Afterwards, each well was pipetted with 40 μL of 0.2 mg/mL *P*-iodonitrotetrazolium violet (INT) dissolved in sterile distilled water. Subsequently, the mixture was incubated for 30 min at 37 °C. Thereafter, MIC was determined as the least concentration of the extract able to inhibit bacterial growth.

#### 2.12.2. Minimum Bactericidal Concentration (MBC)

The agar dilution method was employed to determine the MBC of the SOD(B)ST2SA2 extract [48]. Briefly, from the wells that lacked bacterial growth after incubation, a loopful of culture medium was taken from each well and streaked on sterile nutrient agar plates and subsequently incubated for 12 h at 37 °C. The least concentration of the extract that completely killed *L. monocytogenes* or STEC isolates was regarded as MBC of the extract.

#### 2.12.3. Mechanism of Antibacterial Activity of the Extract

Interactions of the ligands (pentadecanoic acid, diethyl phthalate, and 1-octadecene) against antibacterial target enzymes, D-alanine:D-alanine ligase (DDI), dihydrofolate reductase, and DNA gyrase B were simulated online by AutoDock Vina. Firstly, the 3D structures of DDI, dihydrofolate reductase, and DNA gyrase B were retrieved from the protein database (PDB ID: 2Zdg, 1DIS, and 1KZN, respectively). The structures were then optimized by deleting heteroatoms, water molecules, and other ligands before adding polar hydrogens by Discovery Studio [50]. Thereafter, chemical structures of the ligands (pentadecanoic acid, diethyl phthalate, and 1-octadecene) were obtained from the PubChem database, and their energy was minimized by UCSF chimera [51]. Next, the docking simulations of the ligands and the proteins were executed by AutoDock Vina, and the best-docked conformations were chosen on the basis of their binding scores [52]. Finally, the docked complexes were visualized and analyzed using the Discovery Studio visualizer [52].

## 3. Results

### 3.1. Physicochemical Parameters of the Marine Water

Table 2 shows the physicochemical parameters of marine water from the different beaches that were analyzed in situ. The temperature ranged from 23 °C to 27 °C on the water surface, while the pH of the water was between 6.55 and 7.13. The water pressure ranged from 748.88 to 759.56 mmHg and salinity was 30.81–35.22 PSU. The concentration of the dissolved oxygen (DO) varied between 30.24% and 43.31%. The dissolved solid concentrations ranged from 23.55 to 26.72 g/L on the beaches.

### 3.2. Isolation and Identification of Actinomycetes

In total, 52 presumptive actinomycetes were isolated from the six beaches: the lowest number (1 isolate) was obtained at Alkantstrand beach, whereas the largest number (19 isolates) was found at Salt Rock beach. The majority of the isolates were obtained from the actinomycetes isolation agar and from the heated water samples.

### 3.3. Screening of Antibacterial Activity of Actinomycetes

Out of the 52 obtained actinomycete isolates, only five strains (BLYST2SA3(i), SOD(A)ST2SA2, SOD(B)ST2SA2, SRST1SA3, and TBST2SA1) demonstrated antibacterial activity against the reference and field strains of STEC (Table 3) and L. monocytogenes (Table 4). SOD(B)ST2SA2 was the only strain that exhibited antibacterial activity during the secondary screening (Table 3 and Table 4).

### 3.4. Molecular Identification and Phylogenetic Analysis

Table 5 shows the five identified marine actinomycete isolates: BLYST2SA3(i), SOD(A)ST2SA2, SOD(B)ST2SA2, SRST1SA3, and TBST2SA1. The strain SOD(B)ST2SA2 that showed bioactivity against STEC and *L. monocytogenes* on the primary and secondary screenings belonged to the genus *Nocardiopsis* (Table 5). The strain showed a percentage similarity of 99.93%–99.5% with several strains of *Nocardiopsis dassonvillei,* including XY236 (MT393632.1), A1W5 (MF321787.1), HR10-5 (JN253591.1), HZNU_N_1 (CP022434.1), XY236 (MH432693.1), *N. dassonvillei* subsp. *albirubida* 0Act 405 (KC514117.1), and *N. dassonvillei* subsp. *albirubida* 0Act920 (MG661744.1).

The taxonomic position of SOD(B)ST2SA2 in relation to the other species in the genus *Nocardiopsis* is shown by the phylogenetic tree in Figure 2.

### 3.5. Extraction of the Secondary Metabolites

Four hundred milliliters of the fermented broth medium by N. dassonvillei SOD(B)ST2SA2 yielded crude extracts weighing 3.1 g, 0.81 g, and 0.1 g when extracted by chloroform, methanol, and ethyl acetate, respectively, and the crude extracts had a brownish appearance.

### 3.6. Chemical Composition Analysis of the Chloroform Extract of N. dassonvillei SOD(B)ST2SA2

The GC–MS spectrum of the chloroform extract showed 85 compounds (Appendix A and Table 6). The unsaturated fatty acid (6-octadecanoic acid, Z) was the main constituent (20.6%), followed by 1-octadecene (9.22%), trifluoroacetoxy hexadecane (7.7.82%), pentadecanoic acid (5.86%), and diethyl phthalate (3.43%) (Table 6).

### 3.7. Antibacterial Activity of the Crude Extracts of N. dassonvillei SOD(B)ST2SA2

#### 3.7.1. Agar-Well Diffusion Method

The findings of the in vitro antibacterial activity of the extracts from N. dassonvillei SOD(B)ST2SA2 are presented in Table 7 and Table 8. Among the crude extracts, only the chloroform extract showed inhibition of the growth of bacterial strains. The inhibition zone diameters varied from 0 to 28.67 ± 1.15 mm. The highest inhibition zone diameters, 28.67 ± 1.15 mm (Table 7; Figure 3) and 16.33 ± 1.15 mm (Table 8), were observed against L. monocytogenes ILestanBR363 and the control strains E. coli O157:H7 ATCC 43888, respectively.

#### 3.7.2. MIC and MBC of the Chloroform Extract

The antibacterial performance of the chloroform extract was further assessed by determining its MIC and MBC, and the results are presented in Table 7 and Table 8. The MIC of the extract against all the *L. monocytogenes* strains was 6.25 mg/mL, whereas against STEC it was in a range of 0.78 to 25 mg/mL. The extract had the lowest MIC (0.78 mg/mL) against STEC KRbyGR83, while it possessed the highest MIC value of 25 mg/mL against STEC KEmpBS18 and STEC KEmpBP21 strains (Table 8). The MBC of the extract varied from 25 to >25 mg/mL against the *L. monocytogenes* strains (Table 7), while against the STEC isolates, the extract had an MBC ranging from 3.13 to >25 mg/mL (Table 8).

### 3.8. Antibacterial Activity Mechanism

#### 3.8.1. Binding Scores of the Docked Complexes

The molecular structures of the ligands’ interactions with the target bacterial proteins showed docking energy scores ranging from −7.2 to −5.0 kcal/mol (Table 9). The positive control (ciprofloxacin) possessed relatively lower binding energy scores to the bacterial proteins than the test ligands. Its binding energies ranged from −9.6 to −7.5 kcal/mol. Among the test ligands, diethyl phthalate had lower binding energy scores to the proteins, varying from −7.2 to −5.8 kcal/mol. All the ligands displayed relatively lower binding energy scores to dihydrofolate reductase compared to the other bacterial proteins.

#### 3.8.2. Molecular Interactions

In the present study, the amino acid residue THR A:116 of dihydrofolate reductase (DHFR) interacted with pentadecanoic acid through a conventional hydrogen bond (Figure 4). Alkyl, pi–alkyl, and van der Waals bonds were the other bonds involved in the molecular interaction. The interactions of diethyl phthalate with the DHFR included the conventional hydrogen, carbon hydrogen, van der Waals, pi–pi stacked, alkyl, and pi–alkyl bonds (Figure 4). The molecular interactions of the active site of DHFR against 1-octadecene did not include any hydrogen bonds. However, other bonds participated, namely, van der Waals, alkyl, pi–sigma, and pi–alkyl bonds (Figure 4). The control, ciprofloxacin, interacted with the enzyme DHFR through the following bonds: van der Waals, conventional hydrogen, carbon hydrogen, unfavourable acceptor–acceptor, pi–sigma, pi–pi stacked, alkyl, and pi–alkyl interactions (Appendix A).

Pentadecanoic acid displayed hydrogen bonds with the amino acid residues LYS A:153 and LYS A:116 of D-alanine:D-alanine ligase (DDl) (Appendix A). Van der Waals, pi–sigma, alkyl, and pi–alkyl bonds also participated in the interactions. LYS A:153 formed a hydrogen bond with a carbonyl group of diethyl phthalate (Appendix A). Van der Waals and pi–pi stacked bonds were also formed between diethyl phthalate and DDl (Appendix A). The interaction of DDI with 1-octadecene did not involve any hydrogen bonds; however, van der Waals, pi–sigma, alkyl, and pi–alkyl bonds participated (Appendix A). GLU A:189, LYS A:153, and LEU A:192 bonded ciprofloxacin to the enzyme DDl using the conventional hydrogen bonds, alkyl, pi–alkyl, pi–pi stacked, and van der Waals bonds (Appendix A).

There was a hydrogen bond between the carbonyl group of pentadecanoic acid and VAL A:167 of DNA gyrase B (Appendix A). Additionally, the ligand showed alkyl and van der Waals bond interactions with the protein. The ester oxygen of diethyl phthalate formed a hydrogen bond with ASN A:46 of DNA gyrase B (Appendix A). The other bonds formed were pi–anion, alkyl, pi–alkyl, and van der Waals bonds. Only van der Waals and alkyl bonds participated in the interaction of 1-octadecene with the enzyme (Appendix A). The carbonyl group of ciprofloxacin formed hydrogen bonds with ARG A:76 and ARG A:136 of the enzyme (Appendix A). Moreover, van der Waals, halogen (fluorine), pi–anion, alkyl, and pi–alkyl interactions were also observed.

## 4. Discussion

The marine environments of KwaZulu-Natal Province, South Africa, are relatively underexplored. It is therefore paramount to bioprospect for potential novel antibiotic-producing marine microorganisms such as actinomycetes [63,64]. In the province, there is a lack of studies investigating the antimicrobial potential of marine actinomycetes against *Listeria* species and multidrug-resistant STEC. To address this research gap, our study focused on the isolation and identification of bioactive actinobacteria from the marine environments of KwaZulu-Natal Province.

The diversity and distribution of marine microorganisms are significantly influenced by the physicochemical parameters of seawater [65]. These parameters include temperature, salinity, pH, TDS, DO levels, specific conductivity, and the presence of other chemical compounds. In the present study, the temperature of samples indicated a high probability of the microbial inhabitants of the studied sites being mesophiles. Our findings are similar to those of a previous study by Meena et al. [66] in the marine sediments from Port Blair Bay, India. The pH of the seawater samples was neutral, implying that the microbial inhabitants may be neutrophiles. In contrast, the sampling site (Mangrove forest) was found to be slightly basic (pH 7.68) [67]. In the current study, salinity was high (30.81–35.22 PSU), supporting the survival of halophilic microorganisms and archaea [68]. Comparable salinity ranges (32.3–34.4 PSU) were documented by Muduli et al. [69] from Port Blair Bay, Andaman and Nicobar Islands, India. The DO observed in this study implied that the bacteria inhabiting the beaches are aerobes and/or aero-tolerant. The concentration of TDS in our sampled sites was higher than the reported typical seawater concentration (3–4 g/l) [70]. The water pressure in some of our sampled sites was higher than the atmospheric pressure (750 mmHg), suggesting that most of the microbes occupying those environments are barophiles [71]. Overall, there were statistically significant variations (*p* < 0.05) in water pressure and temperature among some of the beaches. This may cause differences in the microbial community structures of the respective beaches [72]. Furthermore, the physicochemical parameters measured in this study helped in the isolation and cultivation of the actinobacteria in culture conditions that were closest to their natural habitats.

In our present study, only *N. dassonvillei* strain SOD(B)ST2SA2 showed antibacterial potential against the test bacteria during the secondary screening of antibacterial activity. This suggests that this actinobacterial strain produces bioactive compounds that have antibacterial properties, which may be different from those produced by the other strains tested in the study. Similarly, in a different study by Salaria and Furhan [73], out of the 46 presumptive actinomycetes screened for antibacterial activity, only isolate A41 showed promising antibacterial effects during the secondary screening. The isolation of actinobacterial strains closely related to *N. dassonvillei* is significant because *Nocardiopsis* is considered a rare genus of actinomycetes. *Nocardiopsis* strains are ecologically versatile and biotechnologically important [34]. These bacteria are known to produce a wide range of bioactive compounds, including antimicrobial agents, anticancer substances, tumor inducers, toxins, and immunomodulators [34]. Additionally, they secrete various novel extracellular enzymes such as amylases, chitinases, cellulases, inulinases, xylanases, and proteases. The production of these bioactive compounds and enzymes holds potential for biotechnological applications in various fields [34].

The phylogenetic tree indicated that our isolate *N. dassonvillei* SOD(B)ST2SA2 formed a distinct lineage within the *Nocardiopsis* species. Similarly, Tang et al. [74] found their actinomycete isolate (*Haloactinospora alba* strain YIM 90648^T^ [DQ923130]) to form a branch that was separate from that containing species of *Streptomonospora*, *Nocardiopsis*, and *Thermobifida*.

The organic solvent chloroform is relatively less frequently used to obtain anti-infective crude extracts [75]. However, it produced a higher yield of the crude extract than the common organic solvent(s) used both in this study and in another study by Siddharth and Rai [32]. Probably the nature of the target compounds in our study favours being extracted by chloroform. Additionally, in this study, the bioassay results established that the antibacterial agents were in the chloroform extract. In contrast, Okudoh [76] reported ethyl acetate fraction to possess antibacterial compounds in their study.

The antimicrobial potential of the chloroform crude extract was further affirmed by the presence of bioactive compounds with a wide array of reported antimicrobial and anticancer activities, as revealed by the GC–MS analysis (Table 6). To our knowledge, the present study identifies trifluoroacetoxy hexadecane for the first time in the culture extracts of *Nocardiopsis* spp. and of actinomycetes. However, its isomer (4-trifluoroacetoxy hexadecane) is a natural product found in *Streptomyces sparsus* [77].

Crude extracts with MIC values up to 8 mg/mL are regarded as having at least some degree of inhibitory activity [78,79]. Therefore, in the present study the chloroform extract, with MIC values of 6.25 mg/mL against the *L. moncytogenes* strains, possesses some degree of antibacterial activity against the strains. In contrast, a strong antibacterial activity (MIC = 62.5 µg/mL) of a 4-bromophenol derived from *Nocardipsis* sp. SCA21 was reported by Siddharth and Rai [32] against *L. monocytogenes* ATCC 13932 in a previous study.

According to Aligiannis et al. [80], moderate microbial inhibitors are those crude extracts with MIC values ranging between 0.60 mg/mL and 1.50 mg/mL. Therefore, the chloroform extract in our study had moderate antimicrobial activity against some of the multidrug-resistant STEC strains. In a previous study, a crude extract from *Pseudomonas aeruginosa* exhibited a stronger antibacterial activity (MIC = 0.391 mg/mL) against an *E. coli* strain (ATCC 25925) [81].

AutoDock Vina software was used to simulate the interaction of the GC–MS-identified compounds (ligands) and ciprofloxacin against the antibacterial target enzymes (DDl, DNA gyrase B, and DHFR), which are common among bacterial strains. DDl is a bacterial enzyme involved in cell-wall biosynthesis. It catalyzes the formation of UDP-N-acetylmuramoyl pentapeptide, the peptidoglycan precursor [82]. DNA gyrase is a type II topoisomerase that is essential for bacterial DNA replication and transcription [83]. DHFR catalyzes the conversion of dihydrofolate to tetrahydrofolate (THF), which is required for the activity of folate-dependent enzymes and, as a result, is required for DNA synthesis and methylation [84]. These enzymes are therefore important targets for antibacterial drugs.

The affinity of biomolecular interactions and the efficacy of medications are frequently determined using binding free energy. It is defined as the free energy difference between the bound and completely unbound states [83]. Therefore, the lower the value, the more stable the complex formed between the ligand and target protein [85,86]. The substantially lower protein-binding energy scores of diethyl phthalate in our study compared to the other test ligands indicated that it binds to the target proteins more stably. Generally, all the ligands showed potential to interact with the target bacterial proteins and induce antibacterial effects [87,88,89].

Ligand effectiveness to exert antibacterial action is highly dependent on the type of interactions produced with receptors. Intermolecular interactions help to stabilize a ligand/receptor complex, consequently resulting in growth inhibition. Thus, the observed interactions in this study were perceived to have resulted in the inhibition and killing of the tested bacteria [78,86]. The findings were confirmed by Amer et al. [90], who concluded that the highest antimicrobial activity observed in one of their sulfadimidine analogues was likely due to the strong interaction with the binding site of DNA gyrase through the formation of H-bonds with important amino acids (Asp615 and two with Val634). Alkyl, pi–sigma as well as the halogen (fluorine) bonds observed in this study are categorized as covalent bonds. Ligands forming covalent bonds tend to bind permanently to their target sites [86]. Thus, it was concluded that the strong covalent bonds among the ligand–protein complexes resulted in the observed bacterial growth inhibition and/or death.

## 5. Conclusions

The present study revealed Sodwana Beach as a potential source of bioactive and rare actinomycetes. *N. dassonvillei* strain SOD(B)ST2SA2 was isolated for the first time in the marine waters of the KwaZulu-Natal Province. Important bioactive compounds were identified in its crude extract. In addition, the compound trifluoroacetoxy hexadecane was identified for the first time in the extracts of *Nocardiopsis* spp. and actinomycetes. The bacterial strain showed some degree of antibacterial activity against beef isolates (*L. monocytogenes* strains and multidrug-resistant strains of Shiga-toxigenic *E. coli*). The molecular docking results suggested that the antimicrobial compounds within the crude extract may inhibit the bacterial growth through interacting and binding with essential proteins such as DHFR, DDl, and DNA gyrase B. Future studies should include further characterization of *N. dassonvillei* SOD(B)ST2SA2 and the purification and characterization of its bioactive compounds.

## Figures and Tables

**Figure 1 microorganisms-11-02539-f001:**
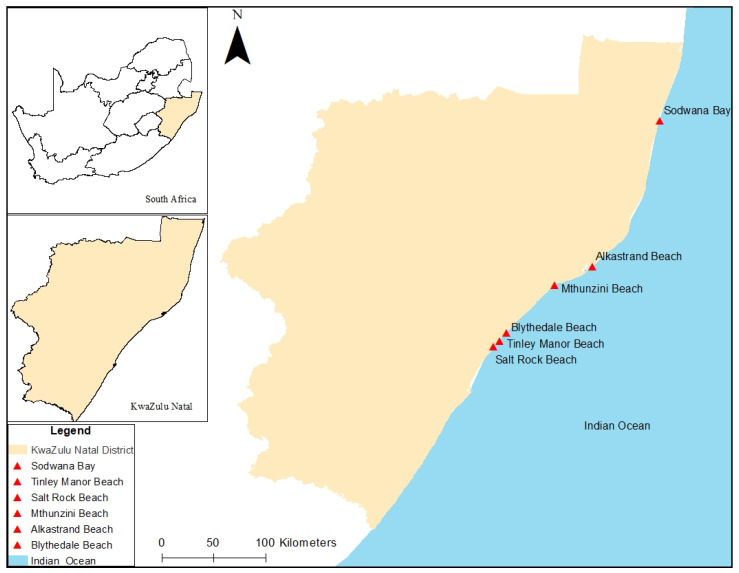
Map showing the six beaches of KZN Province, RSA, where water samples were obtained (Source: Ngema, unpublished).

**Figure 2 microorganisms-11-02539-f002:**
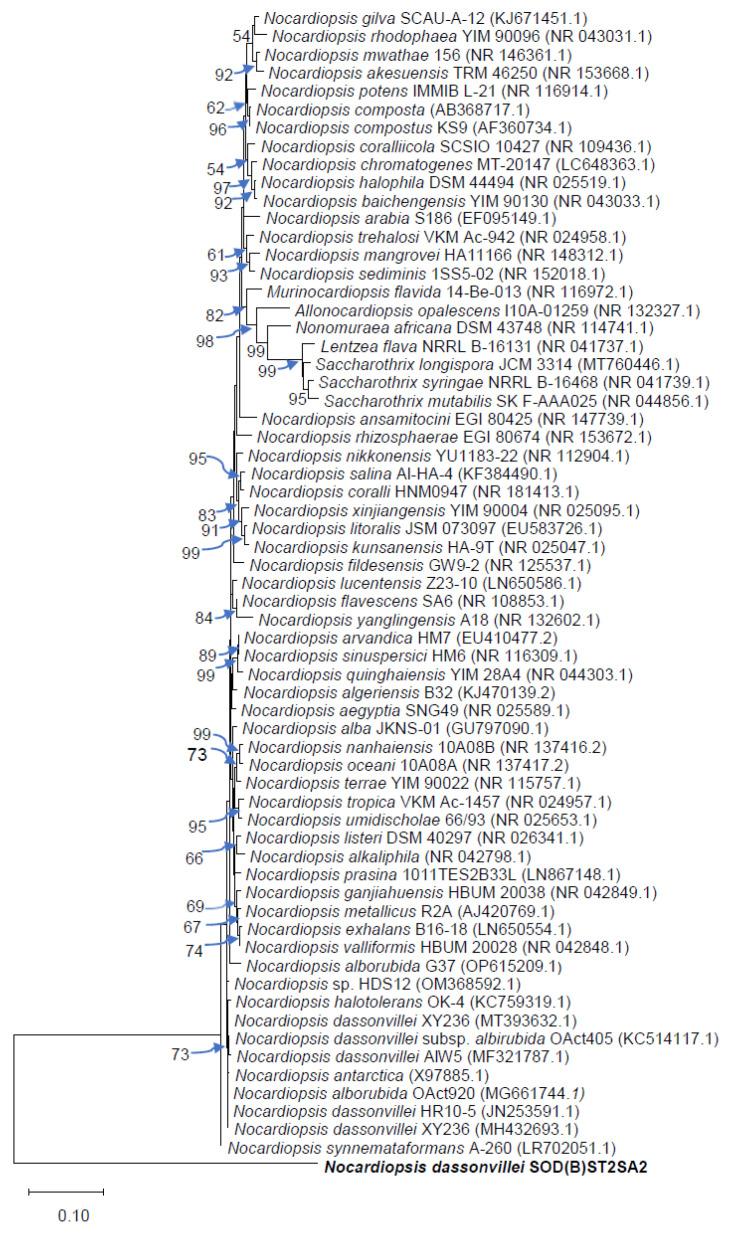
Phylogenetic tree obtained by neighbor-joining analysis of 16S rRNA gene sequences showing the position of N. dassonvillei SOD(B)ST2SA2 and its phylogenetic neighbours. Numbers on branch nodes are bootstrap values (1000 re-samplings; only values over 50% are displayed). Bar, 10% sequence divergence. GenBank accession numbers are also given in parenthesis.

**Figure 3 microorganisms-11-02539-f003:**
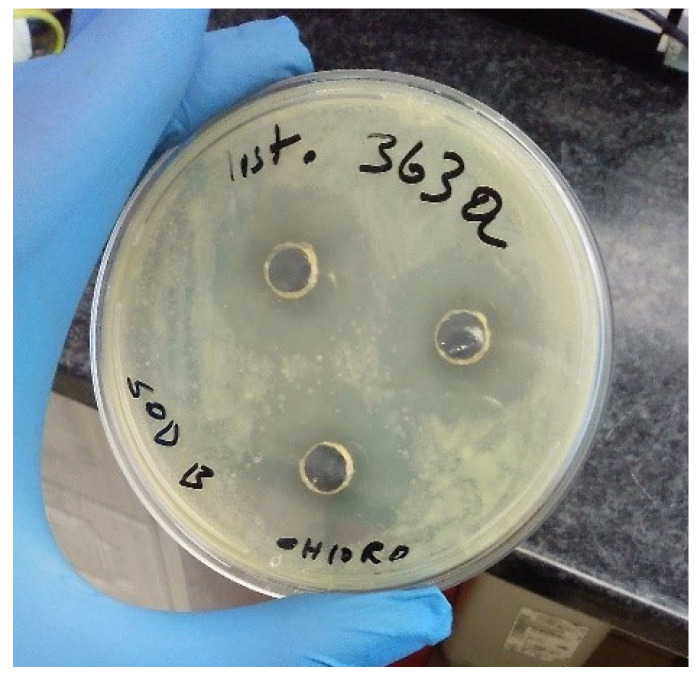
Zones of inhibition (28.67 ± 1.15 mm) of *L. monocytogenes* ILestanBR363 on Mueller–Hinton agar.

**Figure 4 microorganisms-11-02539-f004:**
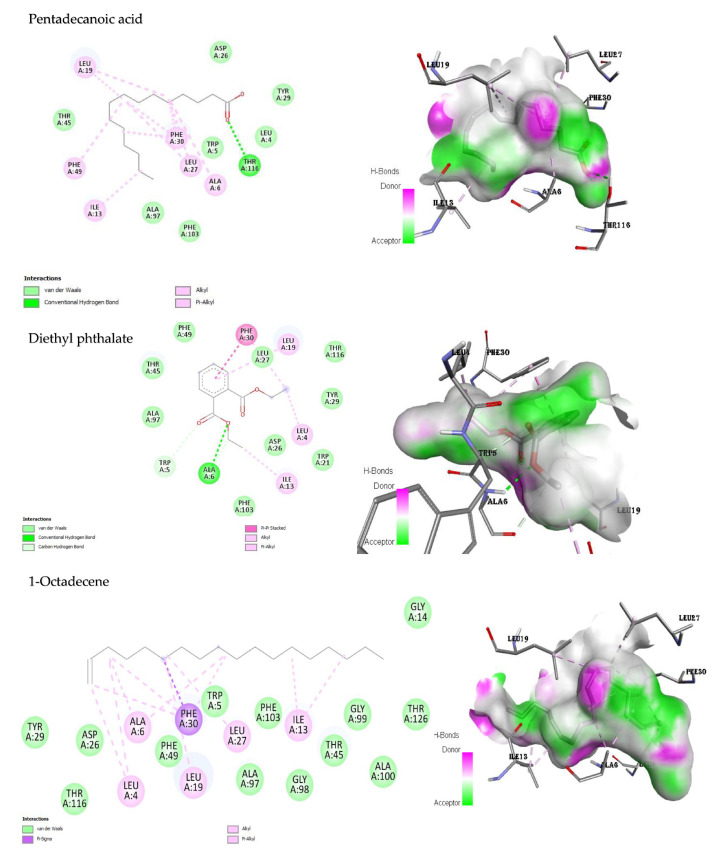
2D and 3D binding interactions of the ligands against dihydrofolate reductase (PDB ID: 1DIS), respectively. The 3D interactions show the ligand in a binding pocket of the enzyme. Dashed lines indicate the interactions between the ligands and the amino acids of the enzyme.

**Table 1 microorganisms-11-02539-t001:** *L. monocytogenes* and STEC isolates from beef and beef products.

Sample Code	Sample Description	*L. monocytogenes* Strains	STEC Strains	Resistance Phenotype for STEC Strains
KEmpBS18	Chuck steak bone	—	KEmpBS18 *	A25-AP10-TE30-TS25
KEmpBP21	Beef burger	—	KEmpBP21 *	A25-AP10-TE30-TS25
KVulFS71	Brisket	—	KVulFS71 *	A25-AP10-CIP5-C30-FOX30-TE30
KRbyGR83	Biltong powder	—	KRbyGR83 *	A25-AP10-CIP5-C30-TS25
KGEO151	Ox intestines	—	KGEO151 *	A25-AP10-TE30-TS25
KGEO161	Ox liver	KGEO161	KGEO161	A25-AP10
KmelDO248	Cow heels	—	KmelDO248 *	A25-AP10-C30-TS25
ILemanEO299	Ox lungs	ILemanEO299	ILemanEO299 *	A25-AP10-C30-TS25
ILemanER317	Burger	ILemanER317	ILemanER317	A25-AP10-FOX30
ILemanAP345	Mince	ILemanAP345	—	
ILestanBR361	Droewors	ILestanBR361	—	
ILestanBR363	Droewors	ILestanBR363	—	
ILestanGP395	Mince	ILestanGP395	—	
ILestanGP400	Burger patties	ILestanGP400	—	

Key: —, indicates that no strains were isolated; *, refers to multidrug-resistant strains; A25, AP10, CIP5, C30, TS25, and FOX30 refer to amoxicillin (25 µg), ampicillin (10 µg), ciprofloxacin (5 µg), chloramphenicol (3 0 µg), co-trimoxazole (1.25–23.75 µg), and cefoxitin (30 µg), respectively.

**Table 2 microorganisms-11-02539-t002:** Physicochemical parameters of marine water on the different beaches of KwaZulu-Natal.

Beaches	Physicochemical Parameters						
	pH ± SD	DO (%) ± SD	Specific Conductivity (mS/cm) ± SD	Total Dissolved Solids (g/L) ± SD	Salinity (PSU) ± SD	Temperature (°C) ± SD	Pressure (mmHg) ± SD
Alkantstrand	6.66 ± 0.22 ^ab^	30.22 ± 4.12 ^a^	49.14 ± 4.71 ^a^	24.57 ± 2.36 ^a^	32.39 ± 3.50 ^a^	24.17 ± 0.09 ^a^	749.79 ± 0.33 ^a^
Blythedale	6.86 ± 0.30 ^ab^	38.70 ± 8.15 ^a^	51.85 ± 0.39 ^a^	25.93 ± 0.19 ^a^	34.11 ± 0.28 ^a^	24.94 ± 0.44 ^a^	748.88 ± 0.10 ^b^
Mthunzini	6.79 ± 0.09 ^ab^	41.28 ± 5.53 ^a^	53.43 ± 0.22 ^a^	26.72 ± 0.11 ^a^	35.22 ± 0.16 ^a^	26.96 ± 0.09 ^b^	754.35 ± 0.16 ^c^
Salt Rock	6.73 ± 0.29 ^ab^	40.67 ± 1.72 ^a^	48.96 ± 5.25 ^a^	24.49 ± 2.63 ^a^	32.04 ± 3.70 ^a^	26.03 ± 0.01 ^c^	754.51 ± 0.09 ^c^
Sodwana	7.13 ± 0.05 ^a^	42.41 ± 5.50 ^a^	47.09 ± 10.09 ^a^	23.55 ± 5.04 ^a^	30.81 ± 7.14 ^a^	23.24 ± 0.49 ^d^	759.56 ± 0.49 ^d^
Tinley Manor	6.55 ± 0.13 ^b^	43.31 ± 4.31 ^a^	50.88 ± 2.16 ^a^	25.45 ± 1.08 ^a^	33.41 ± 1.60 ^a^	24.94 ± 0.27 ^a^	754.53 ± 0.09 ^c^

Values with different letters (a, b, c and d) on the same column are significantly (*p <* 0.05) different. SD represents standard deviation.

**Table 3 microorganisms-11-02539-t003:** Screening of antibacterial activity against multidrug-resistant STEC strains.

STEC	Zones of Inhibition (mm)					
	Actinomycetes					
	Primary Screening					Secondary Screening
	BLYST2SA3(i)	SOD(A)ST2SA2	SOD(B)ST2SA2	SRST1SA3	TBST2SA1	SOD(B)ST2SA2
KEmpBS18	+	+++	++	+	−	+
KEmpBP21	++	++	++	+	+	+
KVulFS71	++	−	−	++	−	+
KRbyGR83	++	++	++	++	−	−
KGEO151	+	++	++	+	+	−
KmelDO248	+	+++	+	++	+	+
ILemanEO299	+	+++	++	++	−	+
O157:H7 ATCC 43888	+++	+++	+++	+++	+	+

Key: Inhibition zone diameter index: + (≤ 29 mm) weak activity, ++ (30–49 mm) moderate activity, +++ (≥50 mm) strong activity, and − denotes no activity.

**Table 4 microorganisms-11-02539-t004:** Screening of antibacterial activity against *L. monocytogenes* strains.

*L. monocytogenes*	Zones of Inhibition (mm)					
	Actinomycetes					
	Primary Screening					Secondary Screening
	BLYST2SA3(i)	SOD(A)ST2SA2	SOD(B)ST2SA2	SRST1SA3	TBST2SA1	SOD(B)ST2SA2
KGEO161	+++	+++	+++	+++	+++	+
ILemanAP345	+++	+++	+++	+++	+++	+
ILemanEO299	+++	+++	+++	+++	+++	+
ILemanER317	+++	+++	+++	+++	+++	+
ILestanBR361	+++	+++	+++	+++	+++	+
ILestanBR363	+++	+++	+++	+++	+++	+
ILestanGP395	+++	+++	+++	+++	+++	+
ILestanGP400	+++	+++	+++	+++	+++	+

Key: Inhibition zone diameter index: + (≤29 mm) weak activity and +++ (≥50 mm) strong activity.

**Table 5 microorganisms-11-02539-t005:** BLAST prediction names of actinomycete isolates.

Beach Name	Strain Number	GenBank Closest Known Species (Accession Number)	Identification
Blythedale	BLYST2SA3(i)	*Streptomyces violaceoruber* (CP0205701) *Streptomyces californicus* (CP070260.1)	*Streptomyces* sp.
Sodwana	SOD(A)ST2SA2	*Nocardiopsis dassonvillei* (MF321787.1) *Nocardiopsis dassonvillei* (MH432693.1) *Nocardiopsis dassonvillei* subsp. *albirubida* (MG661744.1) *Nocardiopsis dassonvillei* (CP022434.1) *Nocardiopsis dassonvillei* (JN253591.1)	*Nocardiopsis dassonvillei*
Sodwana	SOD(B)ST2SA2	*Nocardiopsis dassonvillei* (MT393632.1) *Nocardiopsis dassonvillei* (MH432693.1) *Nocardiopsis dassonvillei* subsp. *albirubida* (MG661744.1) *Nocardiopsis dassonvillei* (CP022434.1) *Nocardiopsis dassonvillei* subsp. *albirubida* (KC514117.1) *Nocardiopsis dassonvillei* (JN253591.1) *Nocardiopsis dassonvillei* (MF321787.1)	*Nocardiopsis dassonvillei*
Salt Rock	SRST1SA3	*Streptomyces* sp. (MH910227.1)	*Streptomyces* sp.
Tinley Manor	TBST2SA1	*Streptomyces* sp. (CP047147.1) *Streptomyces albidoflavus* (MF663704.1)	*Streptomyces* sp.

**Table 6 microorganisms-11-02539-t006:** Major compounds identified in the chloroform extract of *N. dassonvillei* SOD(B)ST2SA2 and their potential biological role.

No.	Molecular Formula	Compound	Area %	Possible Biological Activity of the Identified Compound	Reference
1.	C_18_H_34_O_2_	6-Octadecenoic acid, (Z)	20.94	Antimicrobial, anti-inflammatory, anti-androgenic, cancer-preventative, dermatitigenic hypochlolesterolemic, 5-alpha reductase inhibitor, and anemiagenic insectifuge	[53,54]
2.	C_18_H_36_	1-Octadecene	9.22	Antimicrobial and anticancer activities	[55]
3.	C_18_H_33_F_3_O_2_	Trifluoroacetoxy hexadecane	7.82	Antifungal, anti-oxidant, and anticancer activity	[56,57,58]
4.	C_15_H_30_O_2_	Pentadecanoic acid	5.86	Anticancer, antibacterial, and antifungal activity	[59]
5.	C_12_H_14_O_4_	Diethyl phthalate	3.43	Antibacterial, antifungal, and anticancer activity	[60,61,62]

**Table 7 microorganisms-11-02539-t007:** Antibacterial activity of *N. dassonvillei* SOD(B)ST2SA2′s crude extracts against *L. monocytogenes* strains.

*L. monocytogenes* Strains	Zones of Inhibition (mm)		Chloroform Extract		Ciprofloxacin	
	Crude Extract (100 mg/mL)	Control (2 × 10^−2^ mg/mL)	MIC (mg/mL)	MBC (mg/mL)	MIC (mg/mL)	MBC (mg/mL)
	Chloroform	Ciprofloxacin				
KGEO161	15.33 ± 0.58 0 20.33 ± 0.58 14.67 ± 0.58 25.67 ± 1.15 28.67 ± 1.15 21.33 ± 0.58 23.00 ± 1.00 27.00 ± 0.40	24.7 ± 1.2	6.25	> 25	2.5 × 10^−3^	2.5 × 10^−3^
ILemanAP345		27.0 ± 2.7	6.25	25	1.6 × 10^−4^	1.25 × 10^−3^
ILemanEO299		40.3 ± 2.1	6.25	25	1.6 × 10^−4^	1.25 × 10^−3^
ILemanER317		24.0 ± 1.0	6.25	25	6.3 × 10^−4^	5 × 10^−3^
ILestanBR361		36.3 ± 1.2	6.25	> 25	1.6 × 10^−4^	5 × 10^−3^
ILestanBR363		36.3 ± 1.5	6.25	> 25	6.3 × 10^−4^	2.5 × 10^−3^
ILestanGP395		36.0 ± 5.3	6.25	> 25	1.6 × 10^−4^	2.5 × 10^−3^
ILestanGP400		27.3 ± 1.2	6.25	25	6.3 × 10^−4^	1.25 × 10^−3^
ATCC 15313		42.0 ± 2.7	6.25	25	2.5 × 10^−3^	2.5 × 10^−3^

**Table 8 microorganisms-11-02539-t008:** Antibacterial activity of *N. dassonvellei* SOD(B)ST2SA2′s crude extracts against STEC strains.

STEC Strains	Zones of Inhibition (mm)		Chloroform Extract		Ciprofloxacin	
	Crude Extract (100 mg/mL)	Control (2 × 10^−2^ mg/mL)	MIC (mg/mL)	MBC (mg/mL)	MIC (mg/mL)	MBC (mg/mL)
	Chloroform	Ciprofloxacin				
KEmpBS18	10.67 ± 0.58	46.33 ± 2.31	25	25	3.9 × 10^−5^	3.13 × 10^−4^
KEmpBP21	10.33 ± 0.58	45.33 ± 0.58	25	25	3.9 × 10^−5^	7.8 × 10^−5^
KVulFS71	11.67 ± 0.58	38 ± 2.65	3.13	6.25	3.9 × 10^−5^	7.8 × 10^−5^
KRbyGR83	15.67 ± 1.15	47.33 ± 4.16	0.78	3.13	3.9 × 10^−5^	3.9 × 10^−5^
KGEO151	0	42.33 ± 2.31	—	—	3.9 × 10^−5^	3.9 × 10^−5^
KmelDO248	14.00 ± 1.00	48 ± 1.73	1.57	3.13	3.9 × 10^−5^	3.13 × 10^−4^
ILemanEO299	14.33 ± 0.58	50 ± 5.57	25	>25	3.9 × 10^−5^	7.8 × 10^−5^
O157:H7 ATCC 43888	16.33 ± 1.15	40.67 ± 1.15	1.57	3.13	3.9 × 10^−5^	7.8 × 10^−5^

Key:—means antibacterial test not conducted.

**Table 9 microorganisms-11-02539-t009:** Binding energies of ligands to bacterial proteins (receptors).

Ligands	Receptors	Binding Energy Scores (Kcal/mol)
Pentadecanoic acid	DDI	−5.0
	DNA gyrase B	−5.2
	Dihydrofolate reductase	−6.3
Diethyl phthalate	DDI	−5.8
	DNA gyrase B	−6.0
	Dihydrofolate reductase	−7.2
1-Octadecene	DDI	−4.9
	DNA gyrase B	−4.8
	Dihydrofolate reductase	−6.4
Ciprofloxacin (control)	DDI	−7.5
	DNA gyrase B	−7.6
	Dihydrofolate reductase	−9.6

## Data Availability

The data presented in this study are openly available in National Center for Biotechnology Information (NCBI) with accession numbers OR644492 and OR644493.

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
