# Peer review of "Evaluation of Antimicrobial Activity by Marine Nocardiopsis dassonvillei against Foodborne Listeria monocytogenes and Shiga Toxin-Producing Escherichia coli"

_microorganisms, 2023, doi:10.3390/microorganisms11102539_

Round 1

Reviewer 1 Report

ABSTRACT

Line 16-19. Please simplify the first paragraph, there is redundant information.

KEYWORDS

Please change the keywords, they must be different than the title itself.

INTRODUCTION

·       When Listeria is named, please add sp (Listeria sp)

·       Line 58. Please simplify the scientific names using only “L.” (L. welshimeri, L. seeligeri, L. monocytogenes, and L. ivanovii)

·       Idea in line 63-64 requires citation.

·       Please simplify the idea in lines 67-70. Currently is hard to read and follow.

·       Remove “But” in line 71

·       Rewrite the following idea (While a solution the treatment of listeriosis is antimicrobial chemotherapy [17]”

·       Rewrite the following idea (The actinomycetes, which are Gram-positive bacteria are prolific arsenal of antimicrobials and other bioactive compounds [27]).

·       Please add in line 83 “these bacterial group”

·       Reference is needed in line 97

MATERIALS AND METHODS

·       Please rewrite line 123, pH 7 was used?,

·       Please upgrade the quality of figure 1

·       Line 278, I belive is 10% tween 80, right?

RESULTS

·       Please simplify the idea in lines 314-315.

·       Please simplify the idea in lines 387-391. Remove the excesive repetition of  “Nocardiopsis dassonvillei”. It should be something like this:

·       “With several strains of N. dassonvillei, including XY236 387 (MT393632.1), A1W5 (MF321787.1), HR10-5 (JN253591.1), HZNU_N_1 (CP022434.1), XY236 (MH432693.1); N. dassonvillei subsp. 388 albirubida 0Act 405 (KC514117.1), and N. dassonvillei subsp. albirubida 0Act920 (MG661744.1).”

·       Please rewrite the idea in lines 417-420.

·       Please elaborate the explanation of table 6, since authors add a column with the possible biological activity of the identified molecules.

·       If authors already explained that methanol and ethyl acetate do not worked better in comparison with chloroform in the text, please remove those columns in table 7 and 8

DISCUSSION

Please simplify the paragraph in line 611-617

Author Response

RESPONSES TO REVIEWERS’ COMMENTS

Reviewer 1:

Comment. Line 16-19. Please simplify the first paragraph, there is redundant information.

Response: The statement has been simplified to read as follows: “The emergence of multidrug-resistant pathogens creates public health challenges, prompting a continuous search for effective novel antimicrobials”.

Comment: Keywords: Please change the keywords, they must be different than the title itself.

Response: The keywords have been changed. The new key words are: Marine actinomycetes; Nocardiopsis sp.; Antibacterial potential; Molecular docking; Listeria sp

Comment: INTRODUCTION,  When Listeria is named, please add sp (Listeria sp)

Response: This has been implemented.

Comment:  Line 58. Please simplify the scientific names using only “L.” (L. welshimeri, L. seeligeri, L. monocytogenes, and L. ivanovii)

Response: The scientific names have been simplified

Comment:  Idea in line 63-64 requires citation.

Response: Done

Comment:  Please simplify the idea in lines 67-70. Currently is hard to read and follow.

Response: The idea has been simplified and the statement reads as follows: Among them, serogroup O157 is most frequently associated with serious food poisoning [25]. Recently, the incidence of serious disease due to non-O157 serogroups is increasing [23].

Comment:  Remove “But” in line 71

Response: Done

Comment:  Rewrite the following idea (While a solution the treatment of listeriosis is antimicrobial chemotherapy [17]”

Response: The statement has been rephrased to read as follows: “The treatment of listeriosis is antimicrobial chemotherapy [17]. however, for STEC infections, the use of antibiotics is a debatable matter and generally the use of antibiotics is not recommended.”

Comment:  Rewrite the following idea (The actinomycetes, which are Gram-positive bacteria are prolific arsenal of antimicrobials and other bioactive compounds [27]).

Response: The actinomycetes  are Gram-positive bacteria, and produce antimicrobials and other bioactive compounds [28].

Comment:   Please add in line 83 “these bacterial group”

Response: Done

Comment:   Reference is needed in line 97

Response: Done

MATERIALS AND METHODS 

Comment:   Please rewrite line 123, pH 7 was used?,

Response: The statement was rephrased to “The pH of the medium was adjusted to 7 using 0.1 M HCl and 0.1 M NaOH.

Comment:   Please upgrade the quality of figure 1

Response: The figure has been updated.

Comment:   Line 278, I belive is 10% tween 80, right?

Response: We thank the Reviewer for the comment. Yes, that is correct, and this has been revised accordingly.

RESULTS

Comment:   Please simplify the idea in lines 314-315.

Response: The statement has been rephrased to read as follows: Table 2 shows the physicochemical parameters of marine water from the different beaches that were analysed in situ.

Comment:  Please simplify the idea in lines 387-391. Remove the excesive repetition of  “Nocardiopsis dassonvillei”. It should be something like this:

Response: The idea has been simplified to read as: “The strain showed a percentage similarity of 99.93%–99.5% with several strains of Nocardiopsis dassonvillei, including XY236 (MT393632.1), A1W5 (MF321787.1), HR10-5 (JN253591.1), HZNU_N_1 (CP022434.1), XY236 (MH432693.1), N. dassonvillei subsp. albirubida 0Act 405 (KC514117.1), and N. dassonvillei subsp. albirubida 0Act920 (MG661744.1)”.

Comment:  Please rewrite the idea in lines 417-420.

Response: The statement has been rephrased.

Comment:  Please elaborate the explanation of table 6, since authors add a column with the possible biological activity of the identified molecules.

Response: A column with the possible biological activity of the identified molecules has been added.

Comment:  If authors already explained that methanol and ethyl acetate do not worked better in comparison with chloroform in the text, please remove those columns in table 7 and 8.

Response: The columns have been removed from tables 7 and 8.

DISCUSSION

Comment:  Please simplify the paragraph in line 611-617

Response: The paragraph has been simplified.

Sincerely,

Prof. Evelyn Madoroba

Reviewer 2 Report

This article proposes the isolation 0f some marine actinomycetes from South Africa, the evaluation of their in vitro antimicrobial activity against Listeria and STEC, and the characterization of their mechanisms of action by molecular docking.

The article is complex, well structured.

However, the material and method part has some shortcomings:

- In the  test bacteria section, the authors claim to have previously isolated and characterized the bacteria tested. What is the reference in which this is attested? Please add the reference number.

In Minimum Inhibitory concentration (MIC) of the extract section, the authors state that they used ciprofloxacin, 20 microliters per milliliter, as standard. However in Table 7, the concentration of 10 µg/mL is shown. Two concentrations of the standard were used ?

How was the concentration of Crude extracts chosen, i.e. 100 mg/mL? This is a very high concentration. Was it chosen on the basis of literature references? Please specify.

In NO way I agree with expressing the results in two different units of measurement! The minimum inhibitory concentrations of the standard (ciprofloxacin) are expressed in micrograms/ml and of the tested compounds in milligrams per millilitre. Was this done intentionally to avoid noticing the huge discrepancy between the values of the standard and the extracts? Please use the same unit of measurement.
At such MIC values for the tested extract, the antimicrobial potential is extremely modest, it does not look so promising. 

Author Response

RESPONSES TO REVIEWER 2

Comment. - In the test bacteria section, the authors claim to have previously isolated and characterized the bacteria tested. What is the reference in which this is attested? Please add the reference number

Response. The reference number has been added and the statement reads as follows: “The  test bacteria used in this study were isolated and characterized as part of a project on “Antimicrobial Resistance Among Foodborne Pathogens” with reference number THRIP/22/30/11/2017 (Table 1) (unpublished data)”.

 Comment. In Minimum Inhibitory concentration (MIC) of the extract section, the authors state that they used ciprofloxacin, 20 microliters per milliliter, as standard. However in Table 7, the concentration of 10 µg/mL is shown. Two concentrations of the standard were used ?

Response: We thank the reviewer for this observation. The 10 µg/mL was a typographical error, it is supposed to be 20 µg/mL.  I have changed it to 20 µg/mL. Which is written as 2×10-2 mg/mL in the updated version of the manuscript

Comment. How was the concentration of Crude extracts chosen, i.e. 100 mg/mL? This is a very high concentration. Was it chosen on the basis of literature references? Please specify.

Response: The concentration was determined based on literature. The literature reference is [51] in the revised manuscript.  In the updated version of the manuscript, we have included this reference:

  1. Nxumalo, C.I.; Ngidi, L.S.; Shandu, J.S.E.; Maliehe, T.S. Isolation of endophytic bacteria from the leaves of Anredera cordifolia CIX1 for metabolites and their biological activities. BMC Complement. Altern. Med. 2020, 20, 300.

Comment. In NO way I agree with expressing the results in two different units of measurement! The minimum inhibitory concentrations of the standard (ciprofloxacin) are expressed in micrograms/ml and of the tested compounds in milligrams per millilitre. Was this done intentionally to avoid noticing the huge discrepancy between the values of the standard and the extracts? Please use the same unit of measurement.
At such MIC values for the tested extract, the antimicrobial potential is extremely modest, it does not look so promising. 

Response. We thank the Reviewer for this comment. The units of measurement have been standardized.

Sincerely,

Prof. Evelyn Madoroba
